# Predicting the Politics of an Image Using Webly Supervised Data

**Christopher Thomas**    **Adriana Kovashka**
Department of Computer Science
University of Pittsburgh
Pittsburgh, PA 15213
{chris,kovashka}@cs.pitt.edu

## Abstract

The news media shape public opinion, and often, the visual bias they contain is evident for human observers. This bias can be inferred from how different media sources portray different subjects or topics. In this paper, we model visual political bias in contemporary media sources at scale, using webly supervised data. We collect a dataset of over one million unique images and associated news articles from left- and right-leaning news sources, and develop a method to predict the image's political leaning. This problem is particularly challenging because of the enormous intra-class visual and semantic diversity of our data. We propose a two-stage method to tackle this problem. In the first stage, the model is forced to learn relevant visual concepts that, when joined with document embeddings computed from articles paired with the images, enable the model to predict bias. In the second stage, we remove the requirement of the text domain and train a visual classifier from the features of the former model. We show this two-stage approach facilitates learning and outperforms several strong baselines. We also present extensive qualitative results demonstrating the nuances of the data.

## 1   Introduction

One of the goals of the media is to inform, but in practice, the media also shapes opinions [23, 53, 2, 20, 57, 44]. The same issue can be presented from multiple perspectives, both in terms of the text written in an article, and the visual content chosen to illustrate the article. For example, when speaking of immigration, left-leaning sources might showcase the struggles of well-meaning immigrants, while right-leaning sources might portray the misdeeds of criminal immigrants. The type of topics portrayed is also strong cue for the left or right bias of the source media (e.g. tradition is primarily seen as a value on the right, while diversity is seen as a value on the left [15]).

In this paper, we present a method for recognizing the political bias of an image, which we define as whether the image came from a left- or right-leaning media source. This requires understanding: 1) what visual concepts to look for in images, and 2) how these visual concepts are portrayed across the spectrum. Note that this is a very challenging task because many of the concepts that we aim to learn show serious visual variability within the left and right. For example, the concept of "immigration" can be illustrated with a photo of a border wall, children crying behind bars while detained, immigration agents, protests and demonstrations about the issue, politicians giving speeches, etc. Human viewers account for such within-class variance by generalizing what they see into broader semantic concepts or themes using prior knowledge, deduction, and reasoning.

On the other hand, modern CNN architectures learn by discovering recurring textures or edges representing objects in the images through backpropagation. However, the same objects might appear and be discussed *across* the political spectrum, meaning that the simple presence or absence of objects

is not a good indicator of the politics of an image. Thus, model training may fall into poor local minima due to the lack of a recurring discriminative signal. Further, it is not merely the presence or absence of objects that matters, but rather *how* they are portrayed, often in subtle ways.

In order to capture the visual concepts necessary to predict the politics of an image, we propose a method which uses an auxiliary channel at training time, namely the article text that the image is paired with. Our method contains two stages. In the first one, we learn a document embedding model on the articles, then train a model to predict the bias of the image, given the image and the paired document embedding. To be successful on this task, the model learns to recognize visual cues which complement the textual embedding and suggest the politics of the image-text pair. At test time, we want to recognize bias from images alone, without any article text. Thus, in the second training stage of the model, we use the first stage model as a feature extractor and train a linear bias classifier on top. The article text serves as a type of privileged information to help guide learning.

Since recognizing the right semantic and visual concepts amidst intra-class variance requires large amounts of data, we train our approach on webly supervised data: the only labels are in the form of the political leaning of the source that the image came from. However, for testing purposes, we collect human annotations and test on images where annotators agreed on the label. We experimentally show that our method outperforms numerous baselines on both a large held-out webly supervised test set, and the set of crowdsourced annotations.

We believe that recognizing the political bias of a photograph is an important step towards building socially-aware computer vision systems. Such awareness is necessary if we hope to use computer vision systems to automatically tag or describe images (e.g. for the visually impaired) or to summarize large collections of potentially biased visual content. Social media companies or search engines may deploy such techniques to automatically identify the political bent of images or even entire news sites being spread or linked to. Progress has already been made in this space in other domains. For example, Facebook automatically determines users' political leanings from site activity and pages liked [40]. Other works have studied predicting political affiliation from text [11, 73, 68] or even MRI scans [58]. However, *visual* bias understanding has been greatly underexplored. While some work examines *visual persuasion* [31, 26], none analyzes political leaning as we do.

Our contributions are as follows:

- We propose and make available[1] a very large dataset of biased images with paired text, and a large amount of diverse crowdsourced annotations regarding political bias.
- We propose a weakly supervised method for predicting the political leaning of an image by using noisy auxiliary textual data at training time.
- We perform a detailed experimental analysis of our method on both webly supervised and human annotated data, and demonstrate the factors humans use to predict bias in images.
- We show qualitative results that demonstrate the relationship between images and semantic concepts, and the variability in how faces of the same person appear on the left or the right.

## 2   Related Work

**Weakly supervised learning.**   Our work is in the weakly supervised setting, in the sense that other than noisy left/right labels, our method does not receive information about what makes an image left- or right-leaning. This is challenging because there is significant variety in the type of content that can be left-leaning or right-leaning. Thus, our method needs to identify relevant visual concepts based on which to make its predictions. Recently, weakly supervised approaches have been proposed for classic topics such as object detection [45, 8, 78, 72, 75], action localization [69, 56], etc. Researchers have also developed techniques for learning from potentially noisy web data, e.g. [7]. Also related is work in unsupervised discovery of patterns and topic modeling, e.g. [37, 38, 61, 62, 79, 27, 13, 63, 18]. In contrast to these works, our problem exhibits much larger within-class variance (with left and right being the classes of interest). Unlike objects and actions, the differences between left and right live in semantic space as much as they do in visual space, hence our use of auxiliary training inputs.

**Curriculum learning.**   Also relevant are self-paced and curriculum learning approaches [28, 51, 76, 77, 29]. These attempt to simplify learning by finding "easy" examples to learn with first. We too

employ a type of curriculum learning. We first train a multi-modal classifier to predict bias, using the assumption that the relation between text and bias is more direct. We then leverage this model as a feature extractor by adding an image-only politics classifier on top of it. Thus, our method focuses the model on relevant visual concepts using text.

**Privileged information.** Our method also exploits a similar intuition as privileged information methods [65, 60, 25, 43, 17, 22, 4, 35] that use an extra feature input at training time. These approaches use tied weights [4], computing summary statistics [60, 35], or multitask training [17] to guide learning. The closest such method to ours is [22] which uses an approach trained to predict text embeddings from images. The features are then applied on visual-only data. However, in early experiments we showed directly predicting text embeddings from images is much more challenging on our data because of the many-to-many relationship of images with topics (e.g. image of the White House can be paired with text about Trump's children, border control, LGBT rights, etc.).

**Connecting images and text.** To learn the meaning of the images, we elevate the image representation to a semantic one, by connecting images and text. However, because our texts contain a lot of information not relevant to the image, our main method does not predict text from the image. The latter task has received sustained interest [67, 14, 30, 66, 48, 6, 12, 1, 16, 74] but our domain is unique in that articles that are paired with our images are orders of magnitude longer.

**Visual rhetoric.** Our work also belongs to a recent trend of developing algorithms to analyze visual media and the strategies that a media creator uses to convey a message. [31, 32] analyze the skills and characteristics that a politician is implied to have through a photo, e.g. "competent"; we adapt their method as a baseline in our setting. [49] study differences in facial portrayals between presidential candidates, and [70, 71] examine visual differences between supporters of the left or right. We learn to *generate* faces from the left and right. Further, we examine differences in general images rather than just faces. [26, 74] predict the persuasive messages of advertisements, but persuasion in political images is more subtle. These works are based on careful and expensive human annotations, while we aim to discover facets of bias in a weakly supervised way.

**Bias prediction in language.** Prior work in NLP has discovered indicators of biased language and political framing (i.e. presenting an event or person in a positive or negative light). For example, [54, 3] use carefully designed dictionary, lexical, grammatical and content features to detect biased language, using supervision over short phrases. Others [50, 9, 10, 11, 73, 68] have studied predicting politics from text. In contrast, it is not clear what "lexicon" of biased content to use for images.

## 3 Dataset

Because no dataset exists for this problem, we assembled a large dataset of images and text about contemporary politically charged topics. We got a list of "biased" sources from `mediabiasfactcheck.com` which places news media on a spectrum from extreme left to extreme right. We used [47] to get a list of current "hot topics" e.g. immigration, LGBT rights, welfare, terrorism, the environment, etc. We crawled the media sources that were labeled left/right or extreme left/right for images using each of these topics as queries. After identifying images associated with each keyword and the pages they were on, we used [52] to extract articles. We obtained 1,861,336 images total and 1,559,004 articles total. We manually removed boilerplate text (headers, copyrights, etc.) which leaked into some articles.

### 3.1 Data deduplication

Because sources cover the same events, some images are published multiple times. To prevent models from "cheating" by memorization, all experiments are performed on a "deduplicated" subset of our data. We extract features from a Resnet [24] model for all images. Because computing distances between all pairs is intractable, we use [39] for approximate $k$NN search ($k = 200$). We set a threshold on neighbors' distances to find duplicates and near-duplicates. We determine the threshold empirically by examining hundreds of $k$NN matches to ensure all near-duplicates are detected. From each set of duplicates, we select one image (and its associated article) to remain in our "deduplicated" dataset while excluding all others. If the same image appeared in both left and right media sources, we keep it on the side where it was more common, e.g. one left source and three right sources would result in preserving one of the image-text pairs from the right sources. After removing duplicates, we are left with 1,079,588 unique images and paired text on which the remainder of this paper is based.

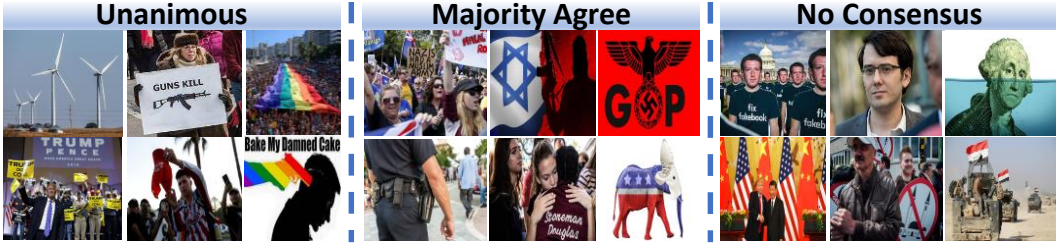

Figure 1: We asked workers to predict the political leaning of images. We show examples here where all annotators agree, the majority agree, and where there was no consensus.

### 3.2 Crowdsourcing annotations

We treat the problem of predicting bias as a weakly supervised task. For training, we assume all image-text pairs have the political leaning of the source they come from. In Sec. 5.3 we show that this assumption is reasonable by leveraging human labels, though it is certainly not correct for all images / text, e.g. a left-leaning source may publish a right-leaning image to critique it. In order to better explore this assumption and understand human conceptions of bias, we ran a large-scale crowdsourcing study on Amazon Mechanical Turk (MTurk). We asked workers to guess the political leaning of images by indicating whether the image favored the left, right, or was unclear. In total, we showed 3,237 images to at least three workers each. We show examples of different levels of agreement in Fig. 1. In total, 993 were labeled with a clear L/R label by at least a majority. We also asked what image features were used to make their guess. The features workers could choose (and the count of each agreed upon) was: closeup-90 (closeup of specific person's face), known person-409 (portrays public figure in political way), multiple people-237 (group or class of people portrayed in political way), no people-81 (scenes or objects associated with parties, e.g. windmill/left, gun/right), symbols-104 (e.g. swastika, pride flag), non-photographic-130 (cartoons, charts, etc.), logos-77 (logo of e.g. CNN, FOX, etc.), and text in image-267 (e.g. text on protest signs, captions, etc.).

We also showed workers the image's article and asked a series of questions about the image-text pair, such as the political leaning of the *pair* (as opposed to image only), the topic (e.g. terrorism, LGBT) the pair is related to, and which article text best aligned with the image. We computed agreement scores and found that 2.45 out 3 annotators agreed on bias label on average, while 1.71 out of 3 agreed on topic, on average. Finally, we asked workers to provide a free-form text explanation of their politics prediction for a small number of images. We extracted semantic concepts from these explanations and later use them to train one of our baseline methods (Sec. 5.1). Humans often mentioned using the positive/negative portrayal of public figures and the gender, race and ethnicity of photo subjects. We provide a demonstration of differences in portrayal across L/R in Sec. 5.5. Absent these cues, workers used stereotypical notions of what issues the left/right discuss or their values. For example, for images of protests or college women, annotators might guess "left".

To ensure quality, we used validation images with obvious bias to disqualify careless workers. We restricted our task to US workers who passed a qualification test, had $\geq 98\%$ approval rate, and who had completed $\geq$1,000 HITs. In total, we collected 14,327 sets of annotations (each containing image bias label, image-text pair bias label, topic, etc.) at a cost of \$4,771. We include a number of experimental results on this human annotated set of images in Sec. 5.3.

## 4 Approach

We hypothesize that the complementary textual domain provides a useful cue to guide the training of our visual bias classifier. The text of the articles includes words that clearly correlate with political bias, e.g. "unite", "medicaid", "donations", "homosexuality", "Putin", "Antifa" and "brutality" strongly correlate with left bias according to our model, while "defend", "retired", "NRA", "minister" and "cooperation" strongly correlate with right bias. By factoring out these semantic concepts into the auxiliary text domain, we enable our model to learn complementary visual cues. We use information flowing from the visual pipeline, and fuse it with the document embedding as an auxiliary source of information. Because we are primarily interested in *visual* political bias, we next remove our model's reliance on textual features, but keep all convolutional layers fixed. We train a linear bias classifier on top of the first model, using it as a feature extractor. Thus, at *test time,* our model predicts the bias of an image *without using any text*. We illustrate our method in Fig. 2.

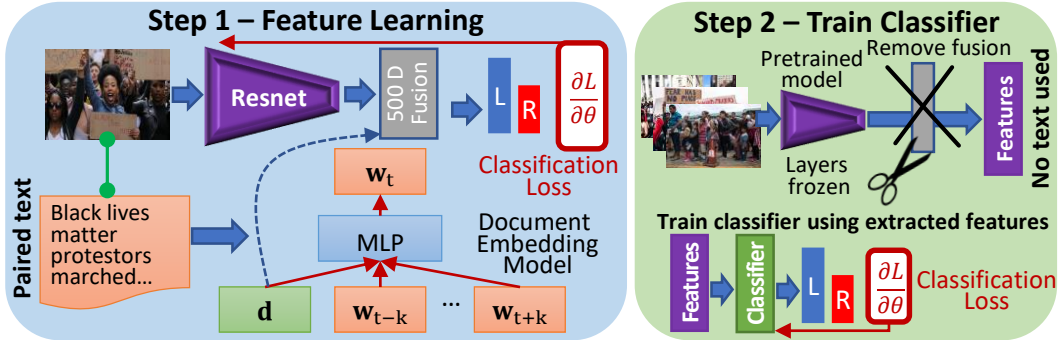

Figure 2: We propose a two-stage approach. In stage 1, we learn visual features jointly with paired text for bias classification. In stage 2, we remove the text dependency by training a classifier on top of our prior model using purely visual features. We show that this approach significantly outperforms directly training a model to predict bias. See Sec. 4.1 for details.

## 4.1 Method details

We wish to capture the implicit semantics of an image by leveraging the association between images and text. More specifically, let

$$\mathcal{D} = \{\mathbf{x}_i, \mathbf{a}_i, \mathbf{y}_i\}_{i=1}^N \tag{1}$$

denote our dataset $\mathcal{D}$, where $\mathbf{x}_i$ represents image $i$, $\mathbf{a}_i$, represents the textual article associated with the $i^{th}$ image, and $\mathbf{y}_i$ represents the political leaning of the image. In the first stage of our method, we seek the following function:

$$f_\theta \left( \mathbf{x}_i, \Omega \left( \mathbf{a}_i \right) \right) = \mathbf{y}_i \tag{2}$$

where $\Omega(.)$ represents transforming the article text into a latent feature space. We train Doc2Vec [36] offline on our train set of articles to parameterize $\Omega$. Specifically, $\Omega$ is trained to maximize the average log probability

$$\frac{1}{T} \sum_{t=1}^{T} \log p \left( \mathbf{w_t} | \mathbf{d}, \mathbf{w_{t-k}}, \ldots, \mathbf{w_{t+k}} \right) \tag{3}$$

where $T$ is the number of words in article $\mathbf{a}$ (we omit the index $i$ to simplify notation), $p$ represents the probability of the indicated word, $\mathbf{w_t}$ is the learned embedding for word $t$ of article $\mathbf{a}$, $\mathbf{d}$ is the learned document embedding of $\mathbf{a}$ (200D), and $k$ is the window around the word to look when training the model. We use hierarchical softmax [42] to compute $p$. We train Doc2Vec on our corpus of news articles, and observe more intuitive embeddings than from a pretrained model.

After training, we compute $\Omega$ for a given article $\mathbf{a}$ by finding the embedding $\mathbf{d}$ that maximizes Eq. 3. $\Omega$ thus projects each article into a space where the resulting vector captures the overall latent context and topic of the article. We provide $\Omega(\mathbf{a})$ to our model's fusion layer for each train image. The fusion layer is a linear layer which receives concatenated image and text features and learns to project them into a multimodal image-text embedding space which is finally used by the classifier.

The formulation of $f_\theta(.)$ described above requires that the *ground-truth* text be available at test time and also does not ensure that our model is learning *visual* bias (i.e. the classifier may be relying primarily on text features and ignoring the visual channel completely). To address this problem, in the second stage of our method, we finetune $f_\theta$ to directly predict the politics of an *image only*, without the text, as follows: $f'_{\theta, \theta'}(\mathbf{x}_i) = \mathbf{y}_i$. Specifically, we freeze the trained convolutional parameters of $f_\theta$ and add a final linear classifier layer to the network, whose parameters are denoted $\theta'$. Because $f_\theta$'s convolutional layers have already been trained jointly with text features, they have already learned to extract visual features which complemented the text domain; we now learn to use those features *alone* for bias prediction, as shown in Fig. 2.

## 4.2 Implementation details

All methods use the Resnet-50 [24] architecture and are initialized with a pretrained Imagenet model. We train all models using Adam [34], with learning rate of 1.0e-4 and minibatch size of 64 images. We use cross-entropy loss and apply class-weight balancing to correct for slight data imbalance between L/R. We use an image size of 224x224 and random horizontal flipping as data augmentation. We use Xavier initialization [21] for non-pretrained layers. We use PyTorch [46] to train all image

models. For our text embedding, we use [55], with $\mathbf{d} \in \mathcal{R}^{200 \times 1}$ and train using distributed memory [36] for 20 epochs with window size $k = 20$, ignoring words which appear less than 20 times.

## 5 Experiments

In this section, we demonstrate our method's performance at predicting left/right bias. We show results on a large held-out test set from our dataset, whose left/right labels come from the leaning of the news source containing the image. We also show results on test images for which a majority of human annotators agreed on the bias and show how humans reason about visual bias. We show that seeing the complementary text information helped *humans* become more accurate at this task, much like seeing the text at training time helps our algorithm. We also show the challenge of our task through across-class nearest-neighbors, how the portrayal of politicians differs from the left to the right, images that best match various words from articles, and visualize how our model makes decisions about visual bias. Our supp. contains additional results such as results per-media source / per-political issue, an exploration of the learned text embedding space, failure cases for machines/humans, humans' reasoning behind their bias decisions, and examples from our dataset.

### 5.1 Methods compared

For quantitative results, we show the accuracy of each method on predicting left/right bias. We compare against the following baselines:

- RESNET [24] - A standard 50-layer classification Resnet.
- JOO [31] - Adaptation of Joo et al.'s method for our task. We use [31]'s dataset to train predictors for 15 attributes and nine "intents" (qualities the photo subject is estimated to have, e.g. trustworthiness, competence). We then use the predictions for these attributes and intents on images from our dataset as additional features to a Resnet to predict a left/right leaning.
- HUMAN CONCEPTS - We use the manually extracted vocabulary of bias-related concepts (e.g. "confederate", "African-American") from the human-provided explanations (Sec. 3.2) and download data for each from Google Image Search. We train a separate Resnet to predict concepts, and use it on each image in our dataset: $p(c_j | \mathbf{x}_i)$ denotes the probability that image $\mathbf{x}_i$ exhibits concept $c_j$. We then use the confidence of each detected concept, as a feature vector to predict bias.
- OCR - We use [41] to recognize free-form scene text in images. Because images contain words not found in the default lexicon (e.g. Manafort), we create our own lexicon from the 100k most common words in our articles. We use [19] for spelling correction. We represent each recognized word as its learned word embedding, denoted $\mathbf{w}_i'$, weighed by the confidence of the recognition $p(\mathbf{w}_i')$ as provided by the recognition model. The feature is thus given by $\frac{1}{n} \sum_{i=1}^{n} p(\mathbf{w}_i') \mathbf{w}_i'$.

All methods use the same residual network architecture. For methods relying on additional features, we use the fusion architecture in Fig. 2. For reference, we also show an upper-bound method OURS (GT) which uses the **G**round **T**ruth text paired with the images *at test time* (to compute a document embedding), in addition to the image. We thus consider it an upper-bound to the task of visual only prediction. OURS (GT) is the same as the first stage of our approach (see Fig. 2, left), without the addition of the image classifier layer in step 2.

### 5.2 Evaluating on weakly supervised labels

In Table 1, we show the results of evaluating our methods on 75,148 held-out images with weakly supervised labels. Our method performs best overall. The top two performing methods rely on semantics discovered in the text domain (OURS and OCR). OCR is unique in that it is able to explicitly use text information at test time, by discovering text within the image and then using word embeddings. OURS improves over OCR by 2.6% (relative 3.8%, reduction in error of 8%). The improvement of OURS over RESNET is 3.4% (relative 5%, error reduction of 11%). This amounts to classifying an additional ~2,555 images correctly. Relying on the concepts humans identified actually slightly *hurt* performance compared to RESNET. This may be because of a disconnect between humans' preconceived notions about L/R and those required by the dataset. We finally observe JOO performs the weakest, likely because [31]'s data mainly features closeups of politicians, while ours contains a much broader image range.

| Method | RESNET | JOO | HUMAN CONCEPTS | OCR | OURS | OURS (GT) |
|---|---|---|---|---|---|---|
| **Accuracy** | 0.678 | 0.670 | 0.675 | 0.686 | **0.712** | 0.803 |

Table 1: Accuracy on weakly supervised labels with the best visual-only prediction method in bold.

| Feature/Method | RESNET | JOO | HUMAN CONCEPTS | OCR | OURS | OURS (GT) |
|---|---|---|---|---|---|---|
| **Closeup** | 0.567 | 0.544 | 0.622 | 0.578 | **0.656** | 0.578 |
| **Known Person** | 0.567 | 0.550 | **0.570** | 0.560 | 0.521 | 0.575 |
| **Multiple People** | 0.722 | 0.671 | 0.688 | 0.730 | **0.768** | 0.705 |
| **No People** | 0.556 | **0.605** | 0.494 | 0.580 | 0.593 | 0.667 |
| **Symbols** | 0.558 | 0.596 | 0.548 | 0.577 | **0.606** | 0.587 |
| **Non-Photographic** | 0.577 | 0.569 | 0.584 | 0.577 | **0.585** | 0.654 |
| **Logos** | 0.545 | 0.584 | 0.597 | **0.662** | 0.623 | 0.584 |
| **Text in Image** | 0.629 | 0.625 | 0.596 | **0.637** | 0.607 | 0.659 |
| **Average** | 0.590 | 0.593 | 0.587 | 0.613 | **0.620** | 0.626 |

Table 2: Accuracy on human consensus labels with the best visual-only prediction method in bold.

## 5.3 Evaluating on human labels

We next tested our methods on test images which at least a majority of MTurkers labeled as having the same bias, i.e. those that humans agreed had a particular label. We describe this dataset in Sec. 3.2. Because workers also labeled images with what features of the image they used to make their prediction, we also break down each method's performance by feature. We show this result in Table 2. OURS performs best on average across all categories and performs best on four out of eight categories. Categories where OURS is outperformed on are reasonable: OCR performs best when text can be relied on in the image, i.e. "logos" and "text in image". We note that while the overall result for OCR approaches OURS, OURS works better on a broader set of images than OCR and is thus a more general method for predicting *visual* bias. OURS is also outperformed by HUMAN CONCEPTS when humans relied on a known face (politician, celebrity, etc.). This may be because HUMAN CONCEPTS relies on external training data (Sec. 5.1) which feature many known individuals, e.g. "rappers" and "founding fathers". JOO outperforms our method when the prediction depends on scene context ("no people"), again likely because JOO uses an external human-labeled dataset to learn features, including scene attributes (e.g. indoor, background, national flag, etc.). We note OURS (GT) performs sig. worse on human labels vs. weakly-supervised labels. This is likely because OURS (GT) has learned to exploit dataset-specific features (e.g. author names, header text, etc.) for prediction, which does not actually translate into humans' commonsense understanding of political bias.

We next test whether our assumption that all images harvested from a right- or left-leaning source exhibit that type of bias is reasonable. Several results computed from our ground-truth human study suggest that our web labels are a reasonable approximation of bias. First, we observe that the relative performance of the methods across Table 1 and 2 is roughly maintained; OURS is best, followed by OCR, and the other methods essentially tied. The results are also sound, e.g. when humans used text, OCR tends to do better, which indicates the model's concept of bias correlates with humans'.

We also performed two other experiments to verify our conclusions. First, we explored the difference between the performance of our method on images on which the *majority* of humans agreed vs. those on which humans *unanimously* agreed. We found that our method worked better when humans unanimously labeled the images vs. simple majority (gain of 4.4%). This suggests that as humans become more certain of bias, our model (trained on noisy data) also performs better. Next, we evaluated the impact of text on humans' bias predictions. We compared how humans *changed* their predictions (made originally using the image only) after they saw the text paired with the image. We found that when workers picked a L/R label, the label was strongly correlated with the weakly supervised label. Moreover, after seeing the text, humans became even more correct with respect to the noisy labels, switching many "unclear" predictions to the "correct" label (i.e. the noisy label). This indicates that: 1) our noisy labels are a good approximation of the true bias of the images; and 2) the paired text is useful for predicting bias (a result also borne out by our experiments).

## 5.4 Quantitative ablations

In order to test the soundness of our method and our experimental design, we performed several ablations. We first tested the importance of the second stage of our method (right side of Fig. 2). To do so, we used OURS (GT), the result of the first stage of our method and instead of performing

| Original | Reconst. | Far left | Far right | Original | Reconst. | Far left | Far right | Original | Reconst. | Far left | Far right |

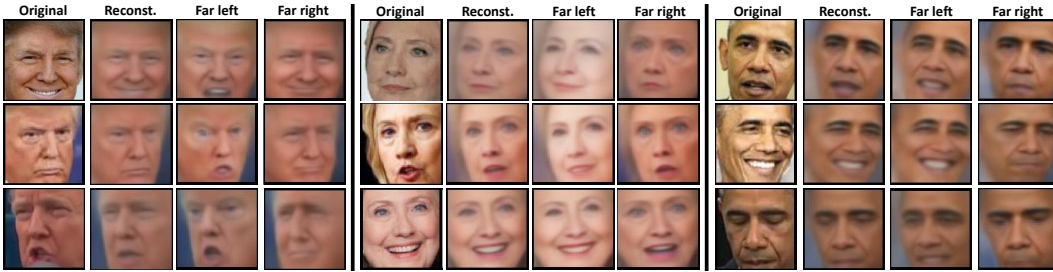

Figure 3: We modified photos to be more left/right. We show the model's "reconstruction" of each face next to the original sample, followed by the sample transformed to the far left and right.

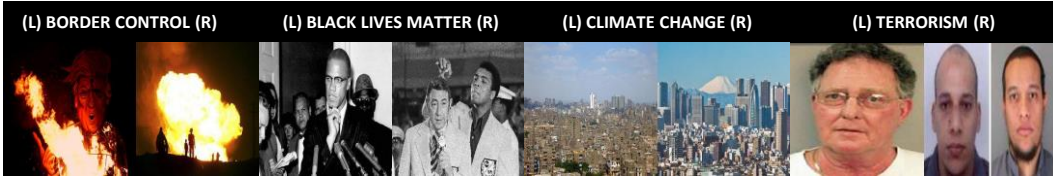

Figure 4: For a set of topics (e.g. LGBT, climate change), we show the closest pair of images across the left/right divide. In each pair, the image on the left is from a left-leaning source, and the one on the right is from a right-leaning source. Note how similar the images in each pair are on the surface.

stage 2, we removed the dependency on text by zeroing out all text embedding weights in the fusion layer. We evaluated on our weakly supervised test set and obtained 0.677, a result sig. worse than our full method, underscoring the importance of stage 2. We next tested how the performance of our method varied given the length of the article text. We thus trained our method with the first $k$ sentences of the article and obtained these results: $k = 1 \rightarrow 0.672$, $k = 2 \rightarrow 0.669$, $k = 5 \rightarrow 0.668$, $k = 10 \rightarrow 0.669$. All choices of $k$ tested performed sig. worse than using the full article (0.712).

We finally examined how reliant our method was on images from a particular media source being in our train set (i.e. to test if the model was learning non-generalizable, source-specific features). We experimented with leaving out all training data harvested from a few popular sources. The result was (before excluding → after excluding): Breitbart (0.607→0.566), CNN (0.873→0.866), CommonDreams (0.647→0.636), DailyCaller (0.703→0.667), DemocraticUnderground (0.713→0.700), NewsMax (0.685→0.628), and TheBlaze (0.746→0.742). We observed only a slight decrease for all sources we tested, suggesting our method is not dependent on seeing the source at train time.

## 5.5 Qualitative results

**Modeling facial differences across politics:** Many workers noted how politicians were portrayed in making their decision (Sec. 3.2). To visualize the differences in how well-known individuals are portrayed within our dataset, we trained a generative model to modify a given Trump/Clinton/Obama face, and make it appear as if it came from a left/right leaning source. We use a variation of the autoencoder-based model from [64], which learns a distribution of facial attributes and latent features on ads, not political images. We train the model using the features from the original method on faces of Trump/Clinton/Obama detected in our dataset using [33]. We use [59] for face recognition. To modify an image, we condition the generator on the image's embedding and modify the distribution of attributes/expressions for the image to match that person's average portrayal on the left/right, following [64]'s technique. We show the results in Fig. 3. Observe that Trump and Clinton appear angry on the far-left/right (respectively) end of the spectrum. In contrast, all three appear happy/benevolent in sources supporting their own party. We also observe Clinton appears younger in far-left sources. In far-right sources, Obama appears confused or embarrassed. These results further underscore that our weakly supervised labels are accurate enough to extract a meaningful signal.

**Nearest neighbors across issues and politics:** In Fig. 4, we show the challenge of classifying in visual space only. We compute the distance between images from the left and right, and show L/R pairs that have a small distance in feature space within topics. For BLM, the left image is serious, while the right image is whimsical. For climate change, one presents a more negative vision, while the other is picturesque. Both border control images show fire, but the left one is of a Trump effigy. For terrorism, the left image shows a white domestic terrorist while the right shows Middle-Eastern men. These pairs highlight how subtle the distinctions between L/R are for some images.

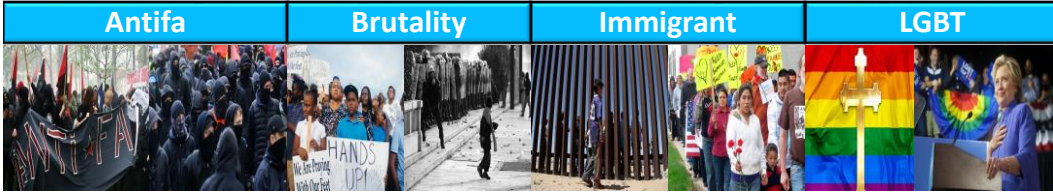

Figure 5: We train a model to predict words from images. The model learns relevant visual cues for each word, demonstrating the utility of exploiting text, even for purely visual classification.

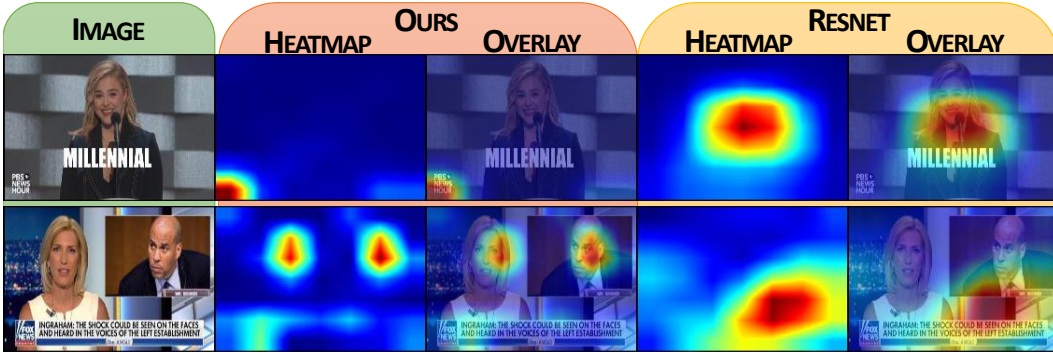

Figure 6: We show visual explanations using [5]. We note that our model looks to logos and faces of public figures, while the baseline uses objects (e.g. mic.) and scene type (e.g. city in background).

**Visualizing image-text alignment:** We wanted to see how well our model could align images and concepts from text. We formulated a variation of our method which, instead of predicting bias, predicted relevant words. We chose a set of 1k words that had the lowest average distance between their images' features (i.e. were visually consistent on avg.) from the 10k most frequent words. The model is trained to predict whether each word is/is not present in the image's article given the image and text embedding. In Fig. 5, we show examples of images that were among the top-100 strongest predictions for that word. We see that the model strongly predicts "antifa" for black-clad protestors, "brutality" for police scenes and protests, "immigrant" for the border wall and Hispanics, and "LGBT" for pride flags. Though the image may only relate to a small portion of the lengthy text, there is enough visual signal present for the model to learn, demonstrating the benefit of leveraging text to complement the model's training.

**Visual explanations:** We wanted to see whether we could interpret how our model learned to perform bias classification. We used Grad-CAM++ [5] to compute attention maps on images that humans annotated. We show the result in Fig. 6. We observe that our model pays the most attention to logos and faces of public figures. We see the model only focuses on the "PBS" logo in the first row (and ignores the face of the lesser known person), but pays attention to both the "Fox News" logo and the face of the well-known commentator in the second row. We believe that because our model was trained with the topic information provided via the text embedding during stage one, the visual component of the model learned to focus on learning visual features that complemented the text (such as logos and faces). Ultimately these features work better even without the text.

## 6 Conclusion

We assembled a large dataset of biased images and paired articles and presented a weakly supervised approach for inferring the political bias of images. Our method leverages the image's paired text to guide the model's training process towards relevant semantics in a way which ultimately improves bias classification. We demonstrate the contribution of our method and dataset both quantitatively and qualitatively, including on a large crowdsourced dataset. Use cases of our work include: inferring the bias of new media sources, constructing balanced "news feeds," or detecting political ads. Broadly speaking, our method demonstrates the potential of using an auxiliary semantic space, e.g. for abstract tasks such as video summarization and visual commonsense reasoning.

**Acknowledgement:** This material is based upon work supported by the National Science Foundation under Grant Number 1566270. It was also supported by an NVIDIA hardware grant. We thank the reviewers for their constructive feedback.

## Footnotes

[1]Our dataset, code, and additional materials are available online for download here: http://www.cs.pitt.edu/~chris/politics

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
