[Supplementary Material 1]

# Predicting the Politics of an Image Using Webly Supervised Data (Supplementary Material)

**Christopher Thomas**    **Adriana Kovashka**
Department of Computer Science
University of Pittsburgh
Pittsburgh, PA 15213
{chris,kovashka}@cs.pitt.edu

## 1   Introduction

This document presents supplementary results to our main text. We first present additional details of our new political bias dataset, in Section 2. Next, in Section 3, we provide two additional quantitative results using our test set which shows the differences between our best performing method and the baselines on the various topics within our dataset. We also provide results for an application of predicting the bias of different media sources. In Section 4, we present additional qualitative results to complement our result in Fig. 5 from the main text, i.e. images that most strongly predicted several words from articles. In Section 5, we illustrate what our trained document embedding model learns by showing nearby words for a number of query words. In Section 6, we compare human vs. machine performance by showing images that either our best algorithm or humans failed to classify (or both). In Section 7, we include additional examples of images agreed upon by human annotators, as well as the free-form text reasons our participants gave for their Left / Right guesses. We also include our MTurk data collection interface. Finally, in Section 8, we show example images and articles from our dataset.

## 2   Dataset Details

In this section, we present additional details of our new political bias dataset to complement our main text. Our dataset contains 1,861,336 images total and 1,559,004 articles total. However, after our deduplication procedure (described in our main text), we are left with 1,079,588 unique images upon which we conduct all experiments. In this section, we break down this *unique* count by politics, topic, and media source. We wish to re-emphasize that even though we exclude duplicates here, the articles associated with duplicate images are not necessarily duplicates (the overwhelming majority are unique). Thus, a large body of potentially useful image-text pairs are excluded from this description because the image associated with the text is not unique.

Figure 1 shows the breakdown of unique images in our dataset by politics. There are more images on the left than on the right, resulting in a slight class imbalance. We correct for this class imbalance during training for all of our experiments by ensuring equal class weight in the loss terms. Figure 2 further breaks down the distribution images by topic. For example, we see our dataset contains 83,145 unique images on the subject of religion (from both L/R), our most frequent category, while we collected 17,073 on the subject of vaccines, our least frequent category.

We also present the frequency distribution of our deduplicated dataset broken down by media source in the attached Microsoft Excel file `media_source_stats.xlsx` as there are too many to include or visualize in this document. Note that we also include the political leaning of the media source, as assigned by Media Bias Fact Check (see our main text for details).

Figure 1: We illustrate the distribution of Left/Right unique images in our deduplicated dataset.

Figure 2: We show the distribution of unique images in our dataset by topic, across both Left/Right.

# 3 Quantitative Results

We present two quantitative results to supplement our main text. We first wanted to understand on what types of images our best performing method, OURS outperformed the RESNET baseline. In Table 1, we show a result which shows the top-3 topics that our method performed the best (and worst) over the baseline. We notice that for no topic does the baseline outperform our method. Even for those topics on which the baseline performs most competitively with our method, our method still outperforms it by 1-2%. We include complete results including additional baselines, for all topics in the included file, `topic_results.xlsx`.

| Method | Vaccines | Fracking | War on Drugs | Border Security | Black Lives Matter | Climate Change |
|--------|----------|----------|--------------|-----------------|--------------------|----------------|
| RESNET | 0.6768 | 0.6737 | 0.6684 | 0.6922 | 0.7026 | 0.6934 |
| OURS | 0.7422 | 0.7209 | 0.7128 | 0.7161 | 0.7269 | 0.7179 |

Table 1: Average performance for the three topics where our method achieves the largest vs smallest improvement over the baseline.

| Method | Top-20 | Top-100 | Sun | Change | Breitbart | NewSt | NewYorker | NatRev | Slate | CNN | RevCom |
|---|---|---|---|---|---|---|---|---|---|---|---|
| RESNET | 0.697 | 0.690 | 0.627 | 0.653 | 0.527 | **0.821** | 0.873 | 0.718 | **0.798** | 0.795 | **0.875** |
| OURS | **0.739** | **0.724** | **0.707** | **0.690** | **0.607** | 0.808 | **0.934** | **0.758** | 0.793 | **0.873** | 0.781 |

Table 2: Average performance for the top-20, and top-100 news sources, and individual results for some popular news sources.

In Table 2, we analyze the results as a function of the media source to which the image belongs. We compute the performance of our method on images exclusively from a particular media source, for each media source. We then rank the sources by number of samples in the test set, and check how performance changes as the number of samples decreases. We see that for media sources with more samples, OURS achieves a stronger result than the RESNET baseline (0.739 vs 0.697). We also show results for individual well-known media sources that have many samples in our dataset. The Sun, Breitbart, and National Review are well-known right-leaning sources, while the rest are left-leaning. Our method works well for both right- and left-leaning sources. For a few left-leaning sources, the baseline achieves stronger results. Among common sources, the baseline's gain is largest on RevCom, a *very* far-left, "revolutionary communism" website. It is surprising to see how accurate we can infer leaning from images alone; close to or over 80% for many sources shown.

We also provide supplementary results to complement this result in `media_source_results.xlsx`, including for other baselines. We break down the performance for each of our methods by media source. We observe that our method, OURS consistently outperforms the baselines, often substantially.

## 4   Image to Word Prediction Results

In our main text, we described a model trained to predict words from images. We trained this model to predict which words, from a fixed dictionary of the 1000 most visual words (see main text for details), would be in the article paired with the image. For this result only, we also conditioned the model on the document embedding of the article paired with the image. After training, we ran our entire large weakly-supervised test set through this model and predicted words for all images. For each word, we then sorted all test set images by the score the model assigned for the prediction of that word and show the 100 images for each that have the highest overall probability. We include results for several words in the `image_to_word` folder. We include results for several words, including "immigrant", "lgbt", "antifa", and "nationalist". We see that the images which strongly predicted the word "immigrant" often feature Hispanic people, children, or law enforcement symbols / personnel. For "lgbt", we notice that many images feature rainbow flags. "Antifa" often features street scenes with protestors wearing black. We also observe fascist symbols, such as swastikas or Nazi salutes in these photos. "Nationalist" features numerous examples of white supremacist imagery, including Ku Klux Klan garbs, swastikas, and Celtic crosses: symbolism associated with white supremacist and neo-nazi movements. Collectively these results indicate that, although the articles paired with the text are lengthly and much more weakly aligned than traditional image-text embedding tasks (i.e. captions, descriptions, etc.), a consistent visual signal exists that our model is able to grasp.

## 5   Textual Embedding Word Retrieval Results

We trained a text embedding [1] model on articles from our dataset. In Table 3 we show an example of what our model learned for a number of query words. We compute the embedding of the query words using our model, then find the nearest words in embedding space from the learned dictionary and rank them. We observe that for "Donald Trump", several of the top words are in Spanish, which are likely coming from articles related to immigration concerning Trump. The translation of these words is fitting in this context, i.e. *intensa* means "intense", while "ultraderecha" means far-right. "Horripilantes" means "horrifying / terrifying." We also notice a "#" sign associated with Trump, likely coming from his use of Twitter. Importantly, we noticed for *events*, like Charlottesville (a protest event in which a protestor was run over by a car in a hate crime), relevant concepts that our *image* classifiers could potentially pick up on appear. For example, "riots", "antifa" (a protest group), "rally", etc. are all visualizable concepts associated with the event. We observe for another event, "Parkland" (a mass school shooting event involving 17 deaths), nearby concepts are "Newtown" (another school shooting), "Hogg" (a survivor of the Parkland shooting), "NRA" (the National Rifle Association, which opposed gun measures following the event and was the subject of significant

| Query phrase: | donald trump | charlottesville | liberal | fascist | parkland |
|---|---|---|---|---|---|
| Results: | auxiliar: 0.4155 | charleston: 0.7303 | leftist: 0.2721 | fascism: 0.7861 | newtown: 0.7640 |
| | intensa: 0.4132 | parkland: 0.7189 | progressive: 0.2650 | fascists: 0.7494 | hogg: 0.7635 |
| | macron: 0.4102 | antifa: 0.7135 | conservative: 0.2583 | nazi: 0.7169 | stoneman: 0.7501 |
| | putin: 0.4042 | kkk: 0.7117 | liberals: 0.2541 | racists: 0.7128 | nra: 0.7455 |
| | ultraderecha: 0.4010 | ferguson: 0.7038 | much: 0.2516 | racist: 0.7068 | charlottesville: 0.7189 |
| | horripilantes: 0.4005 | dallas: 0.6998 | wing: 0.2516 | totalitarian: 0.6903 | shooting: 0.7161 |
| | billionaire: 0.3991 | confederate: 0.6995 | mainstream: 0.2514 | repressive: 0.6866 | walkout: 0.7135 |
| | pence: 0.3980 | richmond: 0.6956 | centrist: 0.2420 | terrorist: 0.6862 | walkouts: 0.7029 |
| | obama: 0.3937 | shooting: 0.6879 | moderate: 0.2323 | filmado: 0.6791 | charleston: 0.7002 |
| | cruz: 0.3928 | horrific: 0.6844 | emerged: 0.2312 | imperialist: 0.6771 | tragedy: 0.6991 |
| | duterte: 0.3924 | portland: 0.6828 | dismal: 0.2309 | communist: 0.6729 | orlando: 0.6986 |
| | erdogan: 0.3919 | riots: 0.6826 | steadily: 0.2269 | nazis: 0.6666 | emma4change: 0.6931 |
| | continuado: 0.3898 | cleveland: 0.6817 | radical: 0.2263 | globalist: 0.6659 | msd: 0.6844 |
| | mueller: 0.3876 | heyer: 0.6806 | portrayed: 0.2256 | nationalist: 0.6655 | sandyhook: 0.6841 |
| | tonight: 0.3874 | protest: 0.6782 | conservatives: 0.2253 | genocidal: 0.6630 | shootings: 0.6795 |
| | inauguration: 0.3869 | rally: 0.6779 | shifted: 0.2248 | rogue: 0.6627 | gun: 0.6752 |
| | gop: 0.3852 | nfl: 0.6760 | defeaning: 0.2245 | authoritarian: 0.6620 | marjory: 0.6739 |
| | america: 0.3848 | tragedy: 0.6757 | plummeted: 0.2244 | extremist: 0.6603 | senseless: 0.6701 |
| | potus: 0.3835 | islamophobia: 0.6727 | outflanked: 0.2219 | vanguard: 0.6599 | kasky: 0.6688 |
| | brexit: 0.3834 | anticom: 0.6721 | progressives: 0.2218 | antifascist: 0.6583 | neveragain: 0.6665 |
| | presidency: 0.3819 | spike: 0.6719 | leftwing: 0.2217 | avakian: 0.6579 | trayvon: 0.6654 |
| | alabama: 0.3817 | berkeley: 0.6718 | touted: 0.2209 | aholes: 0.6571 | 7to: 0.6644 |
| | marcharse: 0.3814 | counterprotesters: 0.6702 | democrat: 0.2209 | waok: 0.6566 | sarasota: 0.6613 |
| | cabinet: 0.3812 | barcelona: 0.6692 | 12,030: 0.2204 | troutdale: 0.6565 | columbine: 0.6610 |
| | netanyahu: 0.3779 | memphis: 0.6679 | long: 0.2196 | clown: 0.6564 | horrific: 0.6597 |
| | milo: 0.3770 | heaphy: 0.6669 | corporatists: 0.2194 | supremacist: 0.6556 | gaskill: 0.6596 |
| | republicans: 0.3766 | alt: 0.6665 | served: 0.2186 | democrat: 0.6548 | manjarres: 0.6596 |
| | opioid: 0.3757 | weekend: 0.6662 | framed: 0.2186 | supremacy: 0.6548 | florida: 0.6583 |
| | comey: 0.3737 | mcauliffe: 0.6657 | hardline: 0.2182 | lunatic: 0.6545 | loesch: 0.6576 |
| | #: 0.3736 | spencer: 0.6654 | leftward: 0.2176 | misogynist: 0.6533 | nationalwalkoutday: 0.6574 |

Table 3: We show examples of our learned text embedding. At the top, we show several "query phrases" which we embed using our method. We then compute the distance from each query phrase to all other learned words in our dataset's vocabulary and rank the words in order of increasing distance. Thus, retrieved words near the top are more closely related to the query phrase in the learned space than words below.

Figure 3: Images that either our best algorithm failed to classify (top), humans failed (bottom left) or both human and machine failed (bottom right). Please see text for our explanation.

press), and a variety of other hashtags and concepts associated with the event. We believe that these results illustrate *how* leveraging text helps our method ultimately perform better by forcing our classifiers to learn to pay attention to certain visual concepts, after being conditioned on the latent document embedding at training time. The representation our image classifiers learn guided by this latent, weak supervision ultimately proves to be superior to the other approaches we tested. We include many additional word query results in `learned_word_embeddings.xlsx`.

## 6 Human vs. Machine Abilities

In Fig. 3, we show images that humans and/or our best-performing algorithm (OURS) were able/unable to classify. At the top, we show the gap between human reasoning abilities and our classifier. The first image at the top has a subtle country vibe (associated with the right), which was imperceptible

for our algorithm. Next is an image of a non-western church, which was likely too different from churches in the training set. The third image shows British actress Emma Thompson campaigning for Greenpeace and getting cited; our algorithm is unable to analyze such complex scenes. At the bottom left of the figure, we show images that humans were unable to classify, but bias in the data helped our algorithm classify. Images of protests, Hollywood, and art, are common in left-leaning images. Finally, we show an image that neither human nor algorithm were able to classify, as it depends on context from the article, which is unavailable at test time.

## 7   MTurk Responses

In Figures 4-6, we show example images which at least a majority (2/3) of humans were able to guess the politics of correctly. We note that many times, when a politician of a particular party is shown, human annotators assume the image is the same leaning as the politician's party (e.g. image of Trump is right). Annotators often rely on racial stereotypes as well ("black women are more liberal," "most rappers are left," "left muslims", '"older white man" is right-leaning). Relying on these stereotypical concepts in our HUMAN CONCEPTS model explains why that model performs best on those images containing humans (see main text), though it doesn't perform best in the dataset at large. We also observe that humans tend to associate the right with guns, patriotic symbols, and religion, whereas they tend to associate peace, compassion, diversity, protests, and minorities with the left. Humans also recognized some of the people appearing in the images and relied on their external knowledge of that person's political leanings to guess the image's label. We also include a complete listing of the concepts that were extracted from the MTurk free-form text explanations in `human_concepts.xlsx`.

We also include our MTurk data collection interface in HTML file `MTurk_Inferface.html`. Note that as you answer the questions, additional questions will appear. We first asked annotators to determine if the image met certain exclusionary criteria, i.e. text, blurry, etc. We then asked annotators to classify the image as left / right / ambiguous. We then asked what features of the image were relied on by the annotator to make their decision. We then showed annotators the article text going with the document and asked whether the text met certain exclusionary criteria, mainly originating from HTML scraping errors. We also asked annotators if the image and text were related to one another and to paste the text from the article that most aligned with the image. We then asked the workers to predict the politics of the image-text *pair* (as opposed to the image alone) and finally asked workers to state political topic(s) of the image-text pair.

## 8   Example Images and Documents from Dataset

In Figures 7-9, we show example images and some text from their associated articles from our dataset. Note that the text we include for each image is truncated, as many of the articles are quite lengthy. We also annotate each image with a "L" or "R" depending on whether the image comes from the left or right respectively, as well as the original source for the image and article text.

We believe these images highlight how extreme some of our media sources are. For example, in Fig. 7, we see an image of apparent Hispanic gang members with Obama's head cropped as one of them. The article discusses illegal immigration and alleges Obama has facilitated allowing "illegals" to stay. In Fig. 8, we see several images of protests, one of which is associated with the left and one of which is associated with the right. We note, however, that the protest image associated with the right (bottom left) actually shows protesters carrying signed *supportive* of Planned Parenthood, a topic associated with the left. A similar situation is found in Fig. 9 where we see an image of a transgender man (bottom row, middle) labeled right. These images' labels only makes sense in conjunction with their paired article text, which are describing circumstances related to the image. These examples underscore one primary challenge of learning visual classifiers on our dataset: *images' labels often depend upon the context on which they appear as much as they depend on what is in the image itself.* Thus, learning to predict the politics from an image is highly challenging due to the inherent high-level semantic nature of the problem as well as the presence of noisy data. We believe our method, guided by privileged information from the text domain helps guide the training and alleviates some of these problems.

| | | | |
|---|---|---|---|
| Republican president<br>Guess: R | A heroic memeified photo of Obama comes form liberals<br>Guess: L | Liberal stance: Anti-discrimination for Hispanics<br>Guess: L | This picture is showing trump supporters at a rally.<br>Guess: R |
| positive picture of Trump<br>Guess: R | A positive picture of Obama<br>Guess: L | Supporting a liberal policy<br>Guess: L | Gun rights supporter are generally right leaning.<br>Guess: R |
| trump smiling<br>Guess: R | PIC OF OBAMA, LIBERAL PRESIDENT<br>Guess: L | Pro immigration<br>Guess: L | Second Amendment shirt would lean right.<br>Guess: R |
| THE LEFT LOVES TO PROTEST.<br>Guess: L | Looks like a man cross dressing so that would only be supported by a left winger<br>Guess: L | many black women are more liberal than conservative<br>Guess: L | the image involves voters and the Republicans are very concerned about the threat of voter fraud<br>Guess: R |
| they like protesting a lot<br>Guess: L | Weirdness embraced<br>Guess: L | Most african american women lean left<br>Guess: L | i chose right because it looks like a voting booth<br>Guess: R |
| Looks like a leftist political rally<br>Guess: L | Looks like a gay person<br>Guess: L | **Guessed incorrectly** | **Guessed incorrectly** |
| the Democratic candidates<br>Guess: L | this attorney is very liberal<br>Guess: L | I believe they are trying to illustrate "police brutality"<br>Guess: L | Obama<br>Guess: L |
| Sanders and Clinton<br>Guess: L | gloria allred is a leftist and represents womens causes<br>Guess: L | seems to be talking about the struggle of african americans<br>Guess: L | Obama is there<br>Guess: L |
| This is from the democratic primary debate<br>Guess: L | **Guessed incorrectly** | **Guessed incorrectly** | For sure left Obama<br>Guess: L |

Figure 4: Examples of images whose politics were correctly guessed by at least a majority (2/3) of MTurkers. We also include the reasons given for their prediction by the MTurkers below each image. MTurkers who guessed the image incorrectly are indicated by "Guessed incorrectly."

# References

[1] Q. Le and T. Mikolov. Distributed representations of sentences and documents. In *International Conference on Machine Learning*, pages 1188–1196, 2014.

| | | | |
|---|---|---|---|
| There's a gun<br>Guess: R | promotes violence<br>Guess: R | shows care and compassion<br>Guess: L | Military leans right.<br>Guess: R |
| guntoting romanticized<br>Guess: R | Flags and knives and who knows what. Seems right leaning.<br>Guess: R | The left mourning, probably another school shooting.<br>Guess: L | military source is often more right<br>Guess: R |
| **Guessed incorrectly** | its an anti-drug image<br>Guess: R | trying to show rallies against current policies<br>Guess: L | **Guessed incorrectly** |
| Right loves the flag<br>Guess: R | older white man<br>Guess: R | This person looks liberal.<br>Guess: L | Lots and lots of guns.<br>Guess: R |
| People on the right tend to cling to national symbols.<br>Guess: R | Military<br>Guess: R | I think this is one of the kids from one of the school shootings.<br>Guess: L | Gun and flag<br>Guess: R |
| **Guessed incorrectly** | Gender and race and flag background.<br>Guess: R | **Guessed incorrectly** | The right would like a more militant George Washington.<br>Guess: R |
| confederate flag<br>Guess: R | Orange popsicle of death<br>Guess: R | African Americans are usualy democrats<br>Guess: L | Looks like a working class white man who is angry with liberals<br>Guess: R |
| Only a person on the right would use a confederate flag.<br>Guess: R | it's not an awful picture of him.<br>Guess: R | African Americans tend to be more to the left.<br>Guess: L | Looks like a right not sure<br>Guess: R |
| confederate flag<br>Guess: R | This is Donald Trump, a republican, though he is making an odd face.<br>Guess: R | Music industry<br>Guess: L | Looks like a white supremist<br>Guess: R |

Figure 5: Examples of images whose politics were correctly guessed by at least a majority (2/3) of MTurkers. We also include the reasons given for their prediction by the MTurkers below each image. MTurkers who guessed the image incorrectly are indicated by "Guessed incorrectly."

| | | | |
|---|---|---|---|
| Blue surroundings<br>Guess: L | left muslims<br>Guess: L | People on the right love church.<br>Guess: R | smiling trump<br>Guess: R |
| I have seen his speeches<br>Guess: L | Minorities tend to fall to the left.<br>Guess: L | Religious windows<br>Guess: R | Trump is shown as happy and giving thumbs up, so this is pro Republican AKA right leaning.<br>Guess: R |
| I think that's Cory Booker, a Democrat hero.<br>Guess: L | **Guesed incorrectly** | **Guesed incorrectly** | President is a Republican<br>Guess: R |
| Looks like a peace rally, a common theme on the left.<br>Guess: L | Screaming woman, must be on the left.<br>Guess: L | Due to the LGBT flag in the street.<br>Guess: L | These are trump supporters<br>Guess: R |
| looks like a candlelight protest.<br>Guess: L | looks like some sort of protest.<br>Guess: L | Definitely left because this is the colors for the gay flag and i am pretty sure that the gay community lean towards left more than the right.<br>Guess: L | because trump is a republican and he would never support a democratic candidate<br>Guess: R |
| fight terroriest<br>Guess: L | the speakers voice.<br>Guess: L | Colors on the road<br>Guess: L | positive image of trump rally<br>Guess: R |
| most rappers are left<br>Guess: L | The images contain solar panels and windmills that are green political<br>Guess: L | I used my instincts and my knowledge on certain people / things.<br>Guess: R | He wants everyone to know he supports LGBT<br>Guess: R |
| Appears to be a liberal which is less conservative and leans to the left.<br>Guess: L | Clean energy<br>Guess: L | photoshopping a giant american flag on a location where it doesn't seem very applicable is just patriotism whoring<br>Guess: R | It is attempting to curry favor for Trump, by showing that he has LGBTQ supporters.<br>Guess: R |
| Freedom of expression<br>Guess: L | **Guessed incorrectly** | It's of the American flag.<br>Guess: R | supportive image of trump<br>Guess: R |

Figure 6: Examples of images whose politics were correctly guessed by at least a majority (2/3) of MTurkers. We also include the reasons given for their prediction by the MTurkers below each image. MTurkers who guessed the image incorrectly are indicated by "Guessed incorrectly."

"She went in thinking that the usual liberal menu of anti-gun policies would reduce that number dramatically. She came out concluding that "the only selling point [of those policies] is that gun owners hate them." That's an interesting way to phrase leftist conventional wisdom in an era when the right's tribalism draws so much scrutiny."….
Source: HotAir.com

"The lower house of the Czech parliament has agreed to alter the constitution so that firearms can be held legally when national security is threatened. The amendment gives Czechs the right to use firearms during terrorist attacks. It was passed by the lower house by a big majority, and is likewise expected to be approved by the upper house."
Source: WND.com

"Former White House chief strategist Steve Bannon told Axios that it's "impossible" that President Donald Trump would pivot to gun control now, warning that such a move would "be the end of everything." Despite taking a gun control stance in the past, Trump knows he got elected on an unambiguous pro-gun stance, and he enjoyed staunch support from the all-powerful NRA. Not even the worst mass killing in U.S. history would be enough to move the president off of that, Trump confidants told Axios."
Source: NewsMax.com

"The lamestream media told you: Illegals at the border have dropped to the lowest point in years, reports The Arizona Republic. The local newspaper reported that 3,100 illegal aliens had come across into Arizona, in a story, late last year. They didn't call them future democrat voters, as some critics claim. Actually, Border Patrol reports 479,371 apprehensions at the border for 2014, nearly a half million, or 39,947 per month on average, most of them in Texas. According to local experts, half a million people is a lot of mouths to feed. It's unclear what happened to them all, or if they'll become future democrat voters as critics claim. Obama has jumped through many hoops to allow many illegals stay in the country, using a pen and a phone, in apparent violation of law."
Source: Ammoland.com

"After a week full of tragedy, death and emotional exhaustion from the American public, a Black Lives Matter rally was quickly planned and hosted in downtown Des Moines this evening. Several young women took the lead in organizing the rally and march, mostly on social media and through personal networks. The result was impressive: around 400 people gathered at Cowles Common at 3rd and Walnut. Only a few blocks from my office, I decided to grab my camera and recorder to walk over after work. I came away from it an hour and a half later confused as to what it had accomplished. "
Source: IowaStartingLine.com

"In 1999, two years before America's longest war would begin in Afghanistan, Lewis Sorley published a seminal work titled A Better War about America's last longest war that raged in the 1960s and 70s. The subtitle of this great work serves as the thesis of the book and says it all: The Unexamined Victories and Final Tragedy of America's Last Years in Vietnam . Effectively beginning when the 88th Congress enacted the Gulf of Tonkin Resolution in August, 1964, which authorized Lyndon Johnson to use military force in …"
Source: RedState.com

Figure 7: Example images and articles (truncated) from our dataset. We annotate each image with the media source from which it and the article came, as well as the politics of that media source, as determined by Media Bias Fact Check (see our main text for details).

"When Jedi Jimenez approached the podium at the People's State of the City Thursday, he faced hundreds of Long Beach residents, crammed shoulder to shoulder in the pews of the First Congregational Church located downtown. Attendees shared one unifying goal: to take their city's issues head on. When Jimenez finally spoke, he didn't just ask for the crowd's attention -- he commanded it. "Over the past year, our country has faced some of the biggest threats to our values of democracy, inclusion and justice."
Source: Daily49er.com

"Like a scout for scholars, the Point Foundation searches out the best and brightest LGBT students. The Los Angeles-based organization grants tens of thousands of dollars in tuition to dozens of collegians each year, some cut off financially from their families because of their sexual orientation. Following a six-month search, the Point Foundation recently announced its 25 scholars of 2010. The diverse group includes a former janitor, a young man who underwent an exorcism at his mother's hands, and a woman, previously fired for being gay, now entering her third year of law school."
Source: Advocate.com

"Two years after disengagement Israel has put a blockade on the Gaza Strip not allowing goods and other necessities into the region making Gazans almost completely aid dependent. Two years ago, Israel completed its unilateral withdrawal from the Gaza Strip. We all remember the intense media campaign shamelessly portraying the settlers as dispossessed victims of a bold move for peace. Among others, Harvard economist Sara Roy argued that Israel's version of disengagement would bring disaster to an already desperate Gaza. Today, we are witnessing emergence of an unparalleled economic catastrophe in the Gaza Strip and with it, the evaporation of the last remaining hopes for a Palestinian state. "
Source: Electronicintifada.net

"On January 23, 2017, the Senate confirmed Rep. Mike Pompeo , a Republican congressman from Kansas, as director of the Central Intelligence Agency. Pompeo, 53, has served in the House of Representatives since 2011. He succeeds a 25-year veteran of the CIA, John Brennan, who's served as the agency's chief since 2013. Advertisement - Continue Reading Below Here's what you need to know about Pompeo: 1. He served in the Army. Mike Pompeo during a TV appearance while he was a member of the Army. Pompeo graduated first in his class from West Point in 1986, according to his congressional biography."
Source: Cosmopolitan.com

"On Sunday, Senator Susan Collins (R-ME) said she would not vote for President Trump's nominee to the Supreme Court if the nominee was "hostile" to Roe v. Wade . This shouldn't come as a surprise; Collins showed how callous she was to the rights of the unborn child in 2003. On October 21, 2003, voting with …"
Source: DailyWire.com

"A Presbyterian chaplain in Maine penned an op-ed this month in a local newspaper in which he described Planned Parenthood as "blessed" and lauded the nation's largest abortion provider for promoting "life-affirming values." The Rev. Marvin Ellison, who ministers to patients at a Planned Parenthood facility in Portland, recently joined with other chaplains to host …"
Source: IJR.com

Figure 8: Example images and articles (truncated) from our dataset. We annotate each image with the media source from which it and the article came, as well as the politics of that media source, as determined by Media Bias Fact Check (see our main text for details).

"Steve Schlariet and Ozzie Russ say they never sought the spotlight of social activism, the spotlight found them. The rural Florida Panhandle couple could become one of the first gay men granted a license to marry in the state when a judge's order takes effect Tuesday. Together nearly 20 years and united during a commitment ceremony in Fort Lauderdale in 2001, the men live a quiet life raising horses and dogs on their central Panhandle ranch. When friends approached them about joining a lawsuit challenging the state's gay marriage ban, Schlariet, 66, and Russ, 48, were a bit reluctant."
Source: LGBTQnation.com

"The Obama administration is currently in the process of considering a series of measures to curb gun violence that would go beyond a ban on assault weapons and high-capacity ammunition, according to the Washington Post . Citing "multiple people involved" in the discussions, the Post says that a working group led by Vice President Joe Biden is considering several sure-to-be controversial measures, such as universal background checks, a system to track weapon sales and …"
Source: Slate.com

"Although the news media and Democrats believe government control of guns owned by Americans is politically necessary, what may be equally important is the investigation into the President Barack Obama-Secretary of State Hillary Clinton illegal weapons deal in Libya that helped to arm the Syrian-based Islamic State of Iraq and Syria (ISIS). The thinking in 2012 was that the fall of the Syrian dictator Bashar al-Assad made a U.S.-Muslim terrorist alliance worth the few negative news stories or Republican…"
Source: ConservativeBase.com

"We are in the midst of Holy Week, a time filled with both highs and lows as we follow Jesus's path from crucifixion to resurrection. In the Christian faith, this is our most sacred occasion. It also serves as an opportunity to spend time with family and loved ones. Sadly, for too many people around the world, Holy Week is a dangerous time. This is especially true for Christians in the Middle East who suffer extreme persecution. In fact, groups like the Islamic State of Iraq and Syria (ISIS) search for and kill Christians simply because of their religious beliefs."
Source: YellowHammerNews.com

"A Catholic theology professor at the College of the Holy Cross in Worcester, Massachusetts is stirring up controversy on campus after a student journalist exposed some of his past writings arguing that Jesus was a "crossdressing," gender-fluid "drag king" who supported gay pedophilia. Senior student Elinor Reilly first wrote about Dr. Tat-siong Benny Liew, who serves as the college's Chair of New Testament Studies, in the college's independent student journal The Fenwick Review. As she explains, Liew has some …."
Source: TownHall.com

"The understandably angry and frustrated student survivors of the deadly school shooting that took place in Parkland,… Fred Guttenberg, father to one of the teens slain in the massacre, confronted Rubio calling his comments in the wake of the shooting "pathetically weak." He called for the senator and the rest of Washington to do something about the gun problem plaguing America, but Rubio's response was about what you would expect, refusal to acknowledge the need for stricter gun laws saying, "the problems we are facing here today cannot be solved by gun laws alone."
Source: TheMaven.net

Figure 9: Example images and articles (truncated) from our dataset. We annotate each image with the media source from which it and the article came, as well as the politics of that media source, as determined by Media Bias Fact Check (see our main text for details).



[Supplementary Material 2]

# Predicting the Politics of an Image Using Webly Supervised Data

Christopher Thomas and Adriana Kovashka

Published in NeurIPS 2019

# OUTLINE

- **Problem introduction**

- Related research

- Dataset

- Our method

- Quantitative results

- Qualitative results

# PREDICTING VISUAL POLITICAL BIAS

- We study predicting the **political leaning of an image**

- Certain political sides are associated with certain demographic groups, concepts, people, etc.
  - We want to see whether we can learn this automatically from the data

- Multimodal setting: images + paired *lengthy* text articles they appeared with
  - We are interested primarily in *visual* bias, not textual

# EXAMPLE IMAGES

?

# OUTLINE

- Problem introduction

- **Related research**

- Dataset

- Our method

- Quantitative results

- Qualitative results

# RELATED RESEARCH – VISUAL PERSUASION

- Visual Persuasion: Inferring Communicative Intents of Images

- Uses facial attributes of known politicians to predict whether the image portrays them in a positive or negative light

- We compare against Joo et al. as a baseline

- In contrast, we don't use human chosen attributes / features; instead we leverage the implicit semantics in the auxiliary text domain to guide training

**Modeling Persuasive Intents**
Joo et al., 2014

Joo, Jungseock, et al. "Visual persuasion: Inferring communicative intents of images." *Proceedings of the IEEE conference on computer vision and pattern recognition*. 2014.

# RELATED RESEARCH – POLITICAL FACES

- Same Candidates, Different Faces: Uncovering Media Bias in Visual Portrayals of Presidential Candidates with Computer Vision

- Looked at 13,026 images from 15 news websites about Clinton / Trump during 2016 election

- Looked at visual attribute differences (e.g., facial expressions, face size, skin condition) between the two candidates

- Used crowdsourced workers to rate a subset of 1,200 images and demonstrated that some visual features also effectively shape viewers' perceptions of media slant and impressions of the candidates
  - **We obtain similar results, but we *generate* faces**

- A big difference between this and our work is we consider images beyond known politicians (we also model these differences generatively)

Peng, Yilang. "Same Candidates, Different Faces: Uncovering Media Bias in Visual Portrayals of Presidential Candidates with Computer Vision." *Journal of Communication* 68.5 (2018): 920-941.

# RELATED WORK – PRIVILEDGED INFORMATION

- Self-supervised learning of visual features through embedding images into text topic spaces

- Uses semantic representation in paired text domain to guide training

- Trains CNN to *predict* latent topics from text, then uses the features from the image model to perform classification

- Our dataset / problem is more challenging because of the **many-to-many** relationship with images to topics (image of White House can be paired with text about immigrants, Trump, Obama, military policy, etc.)
  - Thus, directly predicting text embeddings from image doesn't work as well

Gomez, Lluis, et al. "Self-supervised learning of visual features through embedding images into text topic spaces." *Proceedings of the IEEE Conference on Computer Vision and Pattern Recognition.* 2017.

# OUTLINE

- Problem introduction

- Related research

- **Dataset**

- Our method

- Quantitative results

- Qualitative results

# DATASET COLLECTION

- Used an online resource of biased news sources (from left / right) and politicially contentious issues
  - **20 issues:** Abortion, Black Lives Matter, LGBT, Welfare, etc.

- *Automatically* spidered these sites to find pages with images on them and associated text containing the query phrases

- Extracted **images** and **raw text articles** from the sources
  - Used Dragnet text extraction tool which automatically parses HTML for main article text
  - Process is *noisy*

- Around 1.8M images / articles total

- Dataset is *highly diverse* and also *noisy*

# DATA CLEANUP

- Many news sources report on the same visual content – thus many articles feature the same image

- We extract CNN features for every image in the dataset then we perform approximate KNN search using an off-the-shelf method

- This enables us to find near and exact matches of images

- To form our final dataset, find the side which is most common in the duplicate set and keep one of the instances
  - E.g. 5 times from left, 8 times from right, keep one of the instances from the right and discard all the other instances and their articles

- After cleanup >1M *unique* images and paired articles

# DATASET DETAILS – BREAKDOWN BY POLITICS

Dataset Counts by Politics (after deduplication)

right 41.1%
443979
635609
left 58.9%

# DATASET DETAILS – BREAKDOWN BY ISSUE

Dataset Counts by Issue (after deduplication)

# DATASET CHALLENGES

- Noise in dataset comes from **automatic harvesting**
  - We assume that any images harvested from a left/right site are of that political label, but they actually may be unbiased or have the reverse bias

- Challenges include:
  - Images may be unrelated to query (i.e. unrelated content on page, ads, etc.)
  - Text may fail to parse correctly or contain headers or other noise
  - Lots of noisy images – text, crops of web pages, clipart illustrations, etc.
  - Images that just aren't politically biased

# CROWDSOURCING

- We ran a large-scale crowdsourcing study on Mturk asking workers to guess the political leaning of images
- We showed 3,237 images to at least three workers each
- 993 images were labeled clearly L/R by at least a majority
- We also asked what **image features** workers used to guess
  - E.g. closeup of face, portrays a public figure, a group or class of people is portrayed in a political way, contained symbols (e.g. swastika), etc.
- We also showed workers the article and asked questions about the *pair*
  - What article text is best aligned with the image
  - Topic of the image and article
  - Finally we asked workers to explain their predictions for a small number
- We manually went through the responses and mined concepts used by humans
  - **Recognized people** and used their knowledge + image's portrayal
  - Used **stereotypical concepts** to guess (e.g. African American = Left)
- Queried Google Images for these concepts and trained an image classifier to detect Mturk stereotypical concepts (used as Human Concepts baseline)

| | | | |
|---|---|---|---|
|  |  |  |  |
| Republican president<br>Guess: R | A heroic memeified photo of Obama comes form liberals<br>Guess: L | Liberal stance: Anti-discrimination for Hispanics<br>Guess: L | This picture is showing trump supporters at a rally.<br>Guess: R |
| positive picture of Trump<br>Guess: R | A positive picture of Obama<br>Guess: L | Supporting a liberal policy<br>Guess: L | Gun rights supporter are generally right leaning.<br>Guess: R |
| trump smiling<br>Guess: R | PIC OF OBAMA, LIBERAL PRESIDENT<br>Guess: L | Pro immigration<br>Guess: L | Second Amendment shirt would lean right.<br>Guess: R |

| | | | |
|---|---|---|---|
|  |  |  |  |
| THE LEFT LOVES TO PROTEST.<br>Guess: L | Looks like a man cross dressing so that would only be supported by a left winger<br>Guess: L | many black women are more liberal than conservative<br>Guess: L | the image involves voters and the Republicans are very concerned about the threat of voter fraud<br>Guess: R |
| they like protesting a lot<br>Guess: L | Weirdness embraced<br>Guess: L | Most african american women lean left<br>Guess: L | i chose right because it looks like a voting booth<br>Guess: R |
| Looks like a leftist political rally<br>Guess: L | Looks like a gay person<br>Guess: L | **Guessed incorrectly** | **Guessed incorrectly** |

# CROWDSOURCING CONSENSUS VS NO CONSENSUS

Examples of images where all workers agree, the majority agree, and for which there was no consensus on the left / right leaning

# OUTLINE

- Problem introduction

- Related research

- Dataset

- **Our method**

- Quantitative results

- Qualitative results

# MODEL ARCHITECTURE

- Document embeddings from paired article text act as a source of **privileged information** to help guide training
- Article text is **not** used at test time

- We propose a two-stage approach
- In the first stage, we learn a **document embedding** model from the paired articles
- We then train a Resnet which takes in an image and the document embedding and predicts whether the image-text pair is left/right

# MODEL ARCHITECTURE

- In stage two, we **remove the model's dependency on text**
- We remove the multi-modal fusion layer and train a classifier using the features from the CNN trained in stage 1, while freezing the CNN layers
- Our model thus uses **no text at test time**

# OUTLINE

- Problem introduction

- Related research

- Dataset

- Our method

- **Quantitative results**

- Qualitative results

# EXPERIMENTAL RESULTS – WEAKLY SUPERVISED

| Method | RESNET | JOO | HUMAN CONCEPTS | OCR | OURS | OURS (GT) |
|---|---|---|---|---|---|---|
| Accuracy | 0.678 | 0.670 | 0.675 | 0.686 | **0.712** | 0.803 |

- Accuracy of predicting Left / Right labels on weakly supervised test set
  - Weakly supervised labels are left / right label of the media source the image came from
- Baselines:
  - **Resnet** – An off-the-shelf 50 layer residual network
  - **Joo et al.** – Uses features presented by Joo et al. for predicting visual persuasion + resnet
  - **Human Concepts –** Features of model trained to predict concepts that MTurkers used
  - **OCR –** Resnet + Optical Character Recognition (uses trained word embeddings of detected words)
- *Ours (GT) uses text at test time and is thus not purely a visual prediction*
- **Using text domain to guide training of purely visual model improves performance**

# EXPERIMENTAL RESULTS – HUMAN LABELS

| Feature/Method | RESNET | JOO | HUMAN CONCEPTS | OCR | OURS | OURS (GT) |
|---|---|---|---|---|---|---|
| Closeup | 0.567 | 0.544 | 0.622 | 0.578 | **0.656** | 0.578 |
| Known Person | 0.567 | 0.550 | **0.570** | 0.560 | 0.521 | 0.575 |
| Multiple People | 0.722 | 0.671 | 0.688 | 0.730 | **0.768** | 0.705 |
| No People | 0.556 | **0.605** | 0.494 | 0.580 | 0.593 | 0.667 |
| Symbols | 0.558 | 0.596 | 0.548 | 0.577 | **0.606** | 0.587 |
| Non-Photographic | 0.577 | 0.569 | 0.584 | 0.577 | **0.585** | 0.654 |
| Logos | 0.545 | 0.584 | 0.597 | **0.662** | 0.623 | 0.584 |
| Text in Image | 0.629 | 0.625 | 0.596 | **0.637** | 0.607 | 0.659 |
| Average | 0.590 | 0.593 | 0.587 | 0.613 | **0.620** | 0.626 |

- We also eval. on human labeled data
  - Images that at least a majority of annotators agreed upon

# EXPERIMENTAL RESULTS – HUMAN LABELS

| Feature/Method | RESNET | JOO | HUMAN CONCEPTS | OCR | OURS | OURS (GT) |
|---|---|---|---|---|---|---|
| Closeup | 0.567 | 0.544 | 0.622 | 0.578 | **0.656** | 0.578 |
| Known Person | 0.567 | 0.550 | **0.570** | 0.560 | 0.521 | 0.575 |
| Multiple People | 0.722 | 0.671 | 0.688 | 0.730 | **0.768** | 0.705 |
| No People | 0.556 | **0.605** | 0.494 | 0.580 | 0.593 | 0.667 |
| Symbols | 0.558 | 0.596 | 0.548 | 0.577 | **0.606** | 0.587 |
| Non-Photographic | 0.577 | 0.569 | 0.584 | 0.577 | **0.585** | 0.654 |
| Logos | 0.545 | 0.584 | 0.597 | **0.662** | 0.623 | 0.584 |
| Text in Image | 0.629 | 0.625 | 0.596 | **0.637** | 0.607 | 0.659 |
| Average | 0.590 | 0.593 | 0.587 | 0.613 | **0.620** | 0.626 |

- **Results are sensible**
- **Human Concepts –** Works best on celebrities, politicians, etc.

# EXPERIMENTAL RESULTS – HUMAN LABELS

| Feature/Method | RESNET | JOO | HUMAN CONCEPTS | OCR | OURS | OURS (GT) |
|---|---|---|---|---|---|---|
| Closeup | 0.567 | 0.544 | 0.622 | 0.578 | **0.656** | 0.578 |
| Known Person | 0.567 | 0.550 | **0.570** | 0.560 | 0.521 | 0.575 |
| Multiple People | 0.722 | 0.671 | 0.688 | 0.730 | **0.768** | 0.705 |
| No People | 0.556 | **0.605** | 0.494 | 0.580 | 0.593 | 0.667 |
| Symbols | 0.558 | 0.596 | 0.548 | 0.577 | **0.606** | 0.587 |
| Non-Photographic | 0.577 | 0.569 | 0.584 | 0.577 | **0.585** | 0.654 |
| Logos | 0.545 | 0.584 | 0.597 | **0.662** | 0.623 | 0.584 |
| Text in Image | 0.629 | 0.625 | 0.596 | **0.637** | 0.607 | 0.659 |
| Average | 0.590 | 0.593 | 0.587 | 0.613 | **0.620** | 0.626 |

- **Results are sensible**
- **OCR –** Works best on images containing text in the image

# EXPERIMENTAL RESULTS – HUMAN LABELS

| Feature/Method | RESNET | JOO | HUMAN CONCEPTS | OCR | OURS | OURS (GT) |
|---|---|---|---|---|---|---|
| Closeup | 0.567 | 0.544 | 0.622 | 0.578 | **0.656** | 0.578 |
| Known Person | 0.567 | 0.550 | **0.570** | 0.560 | 0.521 | 0.575 |
| Multiple People | 0.722 | 0.671 | 0.688 | 0.730 | **0.768** | 0.705 |
| No People | 0.556 | **0.605** | 0.494 | 0.580 | 0.593 | 0.667 |
| Symbols | 0.558 | 0.596 | 0.548 | 0.577 | **0.606** | 0.587 |
| Non-Photographic | 0.577 | 0.569 | 0.584 | 0.577 | **0.585** | 0.654 |
| Logos | 0.545 | 0.584 | 0.597 | **0.662** | 0.623 | 0.584 |
| Text in Image | 0.629 | 0.625 | 0.596 | **0.637** | 0.607 | 0.659 |
| Average | 0.590 | 0.593 | 0.587 | 0.613 | **0.620** | 0.626 |

- **Results are sensible**
- **Ours –** Works best on more categories than others and **works best overall**

# OUTLINE

- Problem introduction

- Related research

- Dataset

- Our method

- Quantitative results

- **Qualitative results**

# QUALITATIVE RESULTS

- Trained generative autoencoder on known politicians faces, conditioned on facial semantic attributes / expressions, as well as latent face embedding from autoencoder
- Modify images to be more Left / Right leaning (move embedding towards avg. L/R embedding)
- Trump – Happier on right, angrier/meaner Left
- Hillary – Younger, brighter skin on left, yelling, older on right

# QUALITATIVE RESULTS

- Trained generative autoencoder on known politicians faces, conditioned on facial semantic attributes / expressions, as well as latent face embedding from autoencoder
- Modify images to be more Left / Right leaning (move embedding towards avg. L/R embedding)
- Trump – Happier on right, angrier/meaner Left
- Hillary – Younger, brighter skin on left, yelling, older on right

- Trained generative autoencoder on known politicians faces, conditioned on facial semantic attributes / expressions, as well as latent face embedding from autoencoder
- Modify images to be more Left / Right leaning (move embedding towards avg. L/R embedding)
- Trump – Happier on right, angrier/meaner Left
- Hillary – Younger, brighter skin on left, yelling, older on right

# QUALITATIVE RESULTS

- Trained generative autoencoder on known politicians faces, conditioned on facial semantic attributes / expressions, as well as latent face embedding from autoencoder
- Modify images to be more Left / Right leaning (move embedding towards avg. L/R embedding)
- Trump – Happier on right, angrier/meaner Left
- Hillary – Younger, brighter skin on left, yelling, older on right

- We show closest pair of images across the left/right divide

- Note how similar the images in each pair are on the surface, illustrating the challenge of visual bias prediction

**Query:**

| charlottesville | parkland |
|---|---|
| charleston: 0.7303 | newtown: 0.7640 |
| parkland: 0.7189 | hogg: 0.7635 |
| antifa: 0.7135 | stoneman: 0.7501 |
| kkk: 0.7117 | nra: 0.7455 |
| ferguson: 0.7038 | charlottesville: 0.7189 |
| dallas: 0.6998 | shooting: 0.7161 |
| confederate: 0.6995 | walkout: 0.7135 |
| richmond: 0.6956 | walkouts: 0.7029 |
| shooting: 0.6879 | charleston: 0.7002 |
| horrific: 0.6844 | tragedy: 0.6991 |
| portland: 0.6828 | orlando: 0.6986 |
| riots: 0.6826 | emma4change: 0.6931 |
| cleveland: 0.6817 | msd: 0.6844 |
| heyer: 0.6806 | sandyhook: 0.6841 |
| protest: 0.6782 | shootings: 0.6795 |
| rally: 0.6779 | gun: 0.6752 |

Results

# PREDICTING WORDS FROM IMAGES

| Antifa | Brutality | Immigrant | ALGBT |
|---|---|---|---|

- Train a model to **predict individual words from images** given the image and the document embedding

- The model learns **visual cues for each word**, demonstrating the utility of exploiting text, even for purely visual classification

- Black clad protestors → "antifa", Protestors, police → "Brutality", Border wall / Hispanics → "Immigrant", Pride flags → "LGBT"

# VISUAL EXPLANATIONS

- Our model primarily pays attention to **faces and logos**. The model ignores the face of the person in the first row, but pays attention to the face of the commentator in the second row.

- The model incorrectly predicts the image in the third row; likely because of the logo confuses the model because it likely did not appear in train set and is uncommon

# HUMAN VS. MACHINE ABILITY

We show images that humans and/or our model were able/unable to classify. We note the top left image has a subtle country vibe, while the other two images require familiarity with a non-Western church and Emma Thompson to understand, which our classifier misses. On the bottom left, we see our classifier predicts protests, celebrities, and art as left-leaning. Finally, we show a challenging image that fooled both humans and machine.

# CONCLUSION

- We collected and release a large dataset of biased images and paired article text

- We performed a large-scale human study and collected annotations on our dataset and studied human intuitions surrounding visual political bias

- We presented an approach for predicting the bias of images
  - Uses auxiliary text domain as a source of **privileged information** to guide training

- We showed both quantitative and qualitative experiments demonstrating our method works

- Use cases of our method include automatically inferring bias of media sources or detecting political ads

- Future work may include improved models of image-text alignment, methods for learning joint image-text embedings under noise, and generating biased images