[Reviews · NeurIPS 2019]

Reviewer 1



Originality: This paper proposes a brand new dataset that is unique in that it contains images paired with text with bias labels (both noisy labels from source and also human labels). The methods described are similar to existing distant supervision techniques though they use it for new analysis on this domain. Quality: The experiments seem sound to me. They test on two different sets of labels and achieve consistent results that are reasonable (e.g. ocr performs better with logos). I think there is room for them to add more analysis of the distant supervision technique and possibly include ablation of the approach to verify how much performance gains come from different components. I was curious whether you verified how much performance differs with and without stage 2 of the training? Similarly, is there a “sweet spot” in the amount of text data used helps vs. hurts during training? Clarity: Overall, the paper was well-structured and easy to read, with only a few points that I found confusing. I wanted to clarify about the Ours-GT model described briefly on line 242 because it was a bit ambiguously worded. What do you mean by “ground truth text embeddings”? Significance: The dataset, itself, is a potentially high impact contribution. The techniques and analysis are also interesting to read and indicate possible avenues for future research.

Reviewer 2



1. Problem setting: The problem of predicting political affiliation from news media articles is relevant and important. I am not convinced that the assumption of not having text at test time is a necessary one or even a good one. This assumption is not well motivated by the authors and strongly influences and limits the approach. I view this paper as tackling a real world problem (fairly applied) but unfortunately making strong and unnecessary assumptions to solve it which result in both poor performance (Table 1 shows a gap of 9 points because of this assumption) and an unnecessary two stage approach (Figure 2). A real world application should not throw away information or entire modalities without good reason. 2. Dataset: The dataset collected in this work is original and I do not know of a large dataset containing news media articles and affiliations. From the few qualitative examples of images in the paper, the dataset seems to have a lot of visual variety. 3. Approach: I have made my reservations about the problem setting above. I think the assumption of not having text at test time strongly influences the approach. In the first stage, a model is trained using paired images and text. This uses a ResNet to extract image features, and a Doc2vec model to extract text features. The two features undergo late fusion and are then input to a classifier. In the second stage, a linear classifier is trained only on the fixed ResNet features. The first stage training approach seems to be a standard late fusion method. The second stage, according to me, seems unncessary. Also, training a linear classifier on top of fixed ConvNet features is not uncommon. Questions 1. In L253, the authors say that the JOO method is trained on the closeup of politicians, and thus performs weakest in the 'broader dataset' collected by the authors. In Table 2, however, the JOO method seems to perform the best on "No people". Both these statements don't seem to agree with one another. To add to the confusion, Table 2 also shows that the JOO method's performance on "Closeup" is the methods worst performance (compared to symbols, text etc.). 2. What is the performance of the first stage model? Is it the one denoted by Ours (GT) in Table 1? The results for the Ours (GT) model have not been reported in Table 2. 3. If the assumption of not having text at test time is necessary, the authors should show why their particular style of modeling (two stage) is necessary. How about an approach like DeVISE (Frome et al.)? The ConvNet takes the image as input and has two heads, one to predict the Doc2Vec and the other to predict the political affiliation. Or simply take the top N words in the corpus and then ask the ConvNet to predict those words (along with the political affiliation). 4. What is the "fusion" used in Figure 2? It is never mentioned in the paper. Do you concatenate the features?

Reviewer 3



Originality: There have been other works that look into visual bias for things like advertisements as well as other papers that consider political bias in natural language. The paper does a good job of outlining these works and showing where their model is different. Understanding bias from images seems unique and interesting. Quality: The paper is well cited and put into proper context. There do not appear to be any technical errors in terms of how the model is presented/trained. When splitting the dataset into train/test splits, are individual sources placed into either train or test? E.g., are Breitbart images found in both train and test? If not, could there be a source specific bias which is learned (and not helpful for understanding political bias)? Clarity: The prose is clear and the paper is quite enjoyable to read. There are some important details missing though. In particular, I did not understand how the fusion layer was implemented. I also do not understand what Ours (GT) is in Table 1. It would also be helpful to understand what kinds of errors the bias model is making; the best model in Table 2 is at 62% (chance would be at 50%). What kinds of things does the current model not understand? Results in Table 1 of the supplemental are somewhat helpful, but it would be helpful to know if things like better pose understanding or sentiment analysis would improve results. I see the data collected as a primary contribution of the paper. It might also be helpful for a discussion on how this data could be used by others in the community. Is the intention to have a benchmark task on bias prediction, or are there other aspects of the dataset that would be useful to researchers? It seems like an interesting set of data, but it would be helpful to have this a bit more explicitly outlined. It could also be helpful consider datasheets for datasets for this dataset (Gebru et al. Datasheets for Datasets. Arxiv). It is unclear if the data is biased in such a way that it is not learning about useful/interesting visual bias; in particular, if more right leaning articles discuss gun laws, perhaps the model can learn that any image related to gun laws is right leaning. This is somewhat shown in Figure 3 of the supplemental where (for example) a picture of a person on a red carpet is considered left''. Finally, it would be interesting to also consider politically neutral images. It seems that when collecting the dataset these images were just thrown out. Is this mainly because finding truly neutral images is hard? Understanding if an image is neutral seems important as well. Significance: The significance of this paper comes from the question being asked (can we learn political bias from images?). The collected dataset could be useful for other people interesting in studying bias in images. Additionally, authors include a variety of annotations types (e.g., explanations from humans about decisions) which could be helpful for different types of analysis. The experiments are plentiful. Authors not only consider bias prediction but also image editing to make an image more "left" or "right", accuracy breakdown across different kinds of images, image-text alignment, etc. It would be interesting to see if any sort of interpretability methods could be used to shed light on the results. E.g., when making a prediction what is the most important part of the image for the model to consider? There are no strong claims about their model (is it not novel in comparison to other methods for learning with privileged information?). The paper could be more significant if the model was run on another similar task with good results. I lean towards accept because I think the kinds of questions being asked in this paper are important and would like to encourage more work like this at ML conferences. UPDATE: After reading other reviews, I increased my score to 7.

[Author Response · NeurIPS 2019]

We thank all reviewers for their detailed and constructive comments. **R1:** "potentially high impact contribution,"
"techniques and analysis also interesting," **R2:** "problem [...] is relevant and important," "dataset is original," **R3:** "really
interesting and challenging problem," "prose is clear and the paper is quite enjoyable," "experiments are plentiful".

**All reviewers. Ours-GT:** Apologies for the confusion; we will clarify. OURS-GT uses the **G**round **T**ruth text paired
with the images *at test time* (to compute a document embedding), in addition to the image. We thus consider it an upper
bound to the task of visual only prediction. It is the same as the first stage of our approach, without the addition of the
image classifier layer. We will include results for the upper bound in Table 2 as requested by **R2**.

**R1: Contribution of stage 2:** If we remove stage 2 and zero out weights for text embedding, acc. is only 0.677.

**R1: "Sweet spot" for text data:** We will include an experiment that trains with the first $k$ sentences (varying $k$).

**R2: "Strong assumption" of no text at test time:** As R3 notes, there are numerous works (cited in our main text)
which study the problem of predicting political bias from natural language. In this work, we wanted to explore prediction
of *visual* political bias. We show that predicting bias from images alone is more challenging than prediction from
text. The OURS-GT model uses ground truth text at test time and clearly outperforms all other methods. Our analysis
indicates that certain words and phrases are easy "giveaways" of bias: e.g. "fascist" or "Nazi" imply left bias, while
use of words like "communist" or "socialist" imply right bias. Thus, while a model which has access to test article text
(in addition to the image) performs better at test time, we don't know if the predictions are primarily visual or textual.
From a practical perspective, if one only cares about predicting bias as accurately as possible, they can of course use the
OURS-GT model, or develop novel ways to best utilize the combination of image and text. This is not our focus. The
purpose of our work *from a scientific perspective* is to study purely *visual* political bias. We further show how text can
be leveraged in this context, while still enabling visual-only prediction.

**R2: Problem-specific approach of leveraging text as privileged information:** We refer R2 to Gomez et al., "Self-
supervised learning of visual features through embedding images into text topic spaces", CVPR 2017, which uses
an approach trained to predict text embeddings from images. The features are then applied on visual-only data,
e.g. PASCAL image classification. We tried a variant of this approach of predicting latent text topics from images
and obtained 0.681 on our full dataset (much lower than our method; compare to Tab. 1 in main). Predicting text
embeddings from images is too challenging on our data because of the many-to-many relationship of images w/ topics
(e.g. image of the White House can be paired with text about Trump's children, border control, LGBT rights, etc.).

**R2: Two-headed model / DeVise / Word prediction:** We experimented with a two-headed model which predicted
both bias and the top-1k visual words (see L310 for addl. details) from the first two sentences (these had best overlap
with human chosen aligned text). The model achieved 0.626 acc. for bias prediction overall (much lower than our
method; compare to Tab. 1). Further balancing of loss hyperparameters could potentially improve the result. We
also trained a model that predicted the top 1k visual words from images and then used the word predictions for bias
prediction. This achieved 0.567 acc. Thus predicting words or embeddings is less promising than our approach.

**R2: JOO performance on "No people":** We agree JOO's performance in Tab. 2 is counterintuitive. Note the result
on "No people" is statistically equivalent with our method's performance (McNemar's Test, $p \leq 0.05$). Further, JOO's
method does use scene context features in addition to person features (see L268-269). Finally, JOO's dataset focused on
the 2012 election, while ours primarily deals with 2016, and lacks many of the politicians that appear in ours.

**R3: Per-source train splits:** Because media sources republish images from others for commentary, it is difficult to
ensure *all* images from a source have been excluded. Still, we experimented leaving out all training data harvested from
a few popular sources. The result was (before excluding $\rightarrow$ after excluding): DemocraticUnderground (0.713$\rightarrow$0.700),
DailyCaller (0.703$\rightarrow$0.667), NewsMax (0.685$\rightarrow$0.628), TheBlaze (0.746$\rightarrow$0.742), Breitbart (0.607$\rightarrow$0.566), Common-
Dreams (0.647$\rightarrow$0.636), and CNN (0.873$\rightarrow$0.866). We observed only a slight decrease for all sources we tested.

**R2-R3: Model details / fusion:** The fusion layer is a linear layer which receives concatenated image and text features
and produces activations used by the classifier. Thank you, we will clarify in our main text.

**R3: Failure cases / model interpretability:** We agree that failure cases could be useful and will include some in supp.
We will also include activation heat maps over these images to understand what the model used to make its prediction.

**R3: "Useful" bias:** We agree that the model may be learning things like photo of assault weapon implies left. However,
Figs. 4-6 in supp. show that this is also how *humans* approach this task for ambiguous images (e.g. "guns and flag = R").

**R3: "Neutral" images:** This is an interesting idea, but we consider it orthogonal to our work. Moreover, even agreeing
on a definition of neutral sources is difficult, e.g. people disagree on whether The New York Times is biased.

**R3: Novelty wrt privileged info:** We are not aware of a method like ours. Other approaches use tied weights [4],
computing summary statistics [56, Lambert CVPR 2018], or multitask learning [Elliott IJCNLP 2017].

[Meta-Review · NeurIPS 2019]

The initial scores for this paper were: 8: Top 50% of accepted NeurIPS papers. A very good submission; a clear accept. 3: A clear reject. I vote and argue for rejecting this submission. 6: Marginally above the acceptance threshold. R1 thinks the collected dataset is a potentially high-impact contribution, and also likes the proposed model and results. R2 likes the overall problem set-up, but questions the assumptions made by the proposed model and points to the lack of clarity in the experiments. R3 also thinks the primary contribution is the collected dataset and points to some important missing details. The authors provide a rebuttal. In the post-rebuttal discussion R2 increases their score to 6 as some of their clarity concerns are addressed in the rebuttal but still thinks the model assumptions need further clarification in the paper (see below). R3 increases their score to 7 as many of their concerns were answered in the rebuttal. R1 maintains their positive rating. Given the final positive recommendations 8, 6 and 7, AC recommends accept. AC encourages the authors to further clarify and motivate the model assumptions in the final version of the paper (see also below). Here are anonymised excerpts from the reviewer discussion: R2: “The author response has made the experimental details clear and answers all of my questions satisfactorily. I am still not very convinced of the image only test time assumption. The authors argue this is a scientific assumption rather than a practical one. I am unconvinced this point is made clear in the paper through the motivation or experimental analysis.” R3: “R2 makes an interesting point about whether predicting political bias from *only* images is a realistic assumption. I can imagine scenarios in which people would see an image with little to no text (e.g., while scrolling through social media platforms, I see a lot of news articles with a large image and a fairly short headline). When trying to understand how image bias impacts user clicks or something of that nature, a method like the one described in the paper might be useful.” R1: “I appreciate R2’s point about the limitations of assuming that text is unavailable at test time. While I think this is a fair point, I also can imagine scenarios like the one R3 described. Another potential application for this system is as a resource for people who are interested in doing large-scale image analysis (eg. computational social scientists who want to explore a large set of images for different visual features of political framing, etc.)”